# OpenReview forum: "VISOR++ : VISUAL INPUT BASED STEERING FOR LARGE VISION LANGUAGE MODELS"
_ICLR.cc/2026/Conference — Submitted to ICLR 2026_

### Official Review · Reviewer_w7Jw · 2025-10-27

**Soundness:** 3
**Presentation:** 3
**Contribution:** 3
**Rating:** 6
**Confidence:** 4

**Summary:**

VISOR++ introduces a novel VLM behavioral control method that uses optimized, universal visual inputs to replace traditional activation-based steering vectors. This approach overcomes the limitations of existing techniques (e.g., system prompts being easily overridden, activation steering requiring white-box model access), making behavioral alignment feasible for closed-source and API-served VLMs.Leveraging a differentiable preprocessing pipeline and an adversarial optimization algorithm (CWA-SSA), VISOR++ crafts a single image that effectively steers VLMs like LLaVA and IDEFICS2 across behaviors such as refusal, sycophancy, and survival instinct, all while preserving 99.9% of performance on unrelated MMLU tasks. The research also demonstrates promising directional transferability of these steering images to unseen models (including GPT-4 variants), particularly for negative steering.This work establishes a new, deployment-agnostic paradigm for VLM safety and control, offering a solution with significant universal potential.

**Strengths:**

1. Clear and important motivation. The paper tackles the practical challenge of behavior control for VLMs in safety-critical settings. Prior approaches either rely on brittle system prompts that are easily overridden or on activation-space guidance that requires white-box access. By steering behavior purely through visual input, VISOR++ sidesteps white-box assumptions and is directly applicable to closed-source/API-only models.

2. Methodological novelty. The work shifts guidance from the activation space to the input space by generating adversarially optimized “guidance images.” The differentiable preprocessing pipeline is a key technical contribution, mitigating gradient discontinuities arising from heterogeneous model preprocessing and thereby enabling cross-model optimization.

3. Generalization and transfer. Universal VISOR++ images perform strongly on the integrated model and show preliminary transfer to unseen models (including GPT-4 variants), with particularly consistent directional effects under negative guidance.

**Weaknesses:**

1. Evidence for universal are limited.     While the title/abstract foreground “universality,” the experiments chiefly optimize over a small two-model ensemble (LLaVA-1.5-7B, IDEFICS2-8B).     Transfer to unseen models yields modest Δ effects, particularly weak for positive guidance, with limited impact on Qwen2-VL-7B and Claude Sonnet 3.5.     The universality claim would be more convincing with a larger, more heterogeneous optimization ensemble and stronger transfer results across a broader set of models.     Scaling curves (ensemble size vs. transfer), per-model effect sizes with confidence intervals, and task/prompt diversity would help substantiate the claim.

2. Optimization cost.     Universal image generation relies on double-layer momentum, spectral enhancement, and nontrivial hyperparameter schedules.     Although the method removes runtime access, producing these images still requires gradient access to the integrated model and appears computationally heavy.     For practitioners, the cost and complexity of generating/maintaining universal images may become the new bottleneck.     Reporting wall-clock/GPU-hour budgets, sensitivity to hyperparameters, and simpler ablations (or black-box approximations) would clarify practicality.

3. Case for perturbation results.    The paper does not show representative VISOR++ guidance images or analyze their human perceptibility.     It is unclear whether they are near-imperceptible perturbations or conspicuous textures.     Providing qualitative examples, basic perceptual metrics (e.g., LPIPS/SSIM/PSNR), frequency-domain analyses, and a small human study would illuminate usability, potential covert-attack risks, and deployment considerations.

**Questions:**

1.  Transferability：
Could the authors elaborate on why the transferability for negative steering appears consistently stronger than for positive steering, especially when transferred to unseen models?   Is this asymmetry indicative of fundamental differences in how specific behaviors (e.g., refusal vs. compliance) are represented or aligned within the models' internal states?

2.  Clarifying the mechanism:
CWA-SSA for behavioral steering vs. general adversarial attacks, while the paper leverages adversarial optimization techniques (specifically CWA-SSA), it is crucial to clearly differentiate how VISOR++'s approach uniquely serves 'behavioral steering' rather than merely resulting in simple 'jailbreaks' or 'misclassification' (as often seen in traditional adversarial attacks).   Specifically, given that steering vectors are integrated into the target activations for optimization, how does the underlying mechanism ensure that the generated image steers a broad behavioral tendency across diverse prompts, as opposed to simply maximizing the likelihood of a specific output string or misclassifying a single input?

3.  Visualization : Could the authors please include some typical VISOR++ images in the main paper or appendix?   Furthermore, a discussion on their perceptual properties is needed: are these images subtly perturbed and imperceptible to the human eye, similar to many adversarial examples?

---

> ### Author Response · Authors · 2025-11-23
>
> We thank the reviewers for their feedback. We’re glad you find VISOR++ to be a novel contribution, and we have succeeded in conveying the importance of the current security vulnerabilities, and you find VISOR++ to show consistent results and transferability to other model variants.
>
> __Transferability:__ We refer to the common comment where we clarify how we utilize the terms “universality” and “generalizability”.
>
> __Optimization cost:__ Optimizing 1 image over an ensemble of two models llava and idefics takes 14.4hrs on NVIDIA L40S. We have added this information to the appendix A7.
>
> __CWA-SSA mechanism clarification:__ In our approach, we do not optimize the image with the goal of minimizing a cross-entropy style loss to induce a certain output text. Instead, we optimize the image so that, for a set of prompts from a certain behavioral task, the trained image can minimize the mean-squared-error w.r.t the  activations when steering vectors are applied. This in turn achieves the effect of steering vectors but through the image. Since we utilize a diverse set of prompts using Expectation-over-Transformations, the image is adaptable to new prompts from that distribution. Furthermore, we do it in a fashion where the same image can successfully steer 2 models in the ensemble.
>
>
> __VISOR++ images visualization:__  We have added VISOR++ images to the paper in Figure 3 and Figure 4 in the appendix. We do not see any obvious pattern in the images. Our conjecture is that since the method aims to induce behavioral concepts rather than specific visual concepts, the patterns observed are not visually discernible.

---

> > ### Comment · Reviewer_w7Jw · 2025-11-27
> >
> > Thank you to the authors for your thorough responses.These have addressed most of my initial concerns. However, after considering reviewer EZDE’s comments, I find that most of their points are valid. As a result, I am keeping my original score and adjusting my confidence level to 2.

---

> > > ### Author Response · Authors · 2025-11-30
> > >
> > > We are glad to have addressed most of your initial concerns. We have also included a detailed rebuttal that includes results from all of the ablation studies suggested by reviewer EZDE. From these results, we clearly demonstrate the necessity of dual-momentum with spectral features to provide the maximum steering range.

---

### Official Review · Reviewer_EZDE · 2025-10-27

**Soundness:** 2
**Presentation:** 1
**Contribution:** 2
**Rating:** 2
**Confidence:** 4

**Summary:**

The paper proposes VISOR++, a method for steering the behavior of Vision-Language Models through optimized visual inputs instead of activation-based vectors. By leveraging adversarial optimization with spectral augmentation and dual-momentum updates, VISOR++ can generate universal images that induce specific behavioral shifts (e.g., refusal, sycophancy) across multiple models without accessing their internals. Experiments on LLaVA-1.5 and IDEFICS2 demonstrate comparable steering performance to activation-based methods while maintaining normal task accuracy.

**Strengths:**

1.The paper introduces a novel concept of visual input-based behavioral steering, shifting control from activation space to input space for VLMs.

2.VISOR++ offers a practical, deployment-agnostic approach for behavioral control, potentially useful for closed-source or API-based multimodal systems.

**Weaknesses:**

1.The paper introduces several algorithmic components (e.g., spectral augmentation, dual-momentum optimization) but provides no ablation to isolate their contributions.

2.The transferability study (Table 2) only compares VISOR++ with random images, making it hard to judge relative effectiveness.

3.Baseline comparison is limited to steering vectors and system prompts, omitting relevant visual or multimodal steering methods such as ASTRA or SteerVLM.

3.The abstract incorrectly describes MMLU as containing “14,000 tasks,” while it actually has 57 tasks and about 14 k samples.

4.The paper inconsistently mixes the singular and plural forms of “VLM”/“VLMs” and “LLM”/“LLMs,” which may confuse readers and reflects insufficient attention to writing consistency.

5.Several standard components and models (e.g.,LLaVA, Mistral, Qwen2-VL, LLaVA-NeXT, Llama, etc.) are mentioned without proper citations.

**Questions:**

1.Could you provide an ablation showing VISOR++ performance without the spectral-domain augmentation?

2.How much improvement does the dual-momentum optimization provide over single-momentum or PGD-only optimization?

3.What experimental evidence supports the claim that “PGD is very effective” (line 225)?

---

> ### Author Response · Authors · 2025-11-23
>
> Thank you for your feedback. We’re glad to know that you find VISOR++ to be novel, practical and deployment agnostic. These were points that were very important to us during development, and we are pleased to know they are reflected in the paper.
>
> __Ablations Spectral component:__ We replace spectral domain gradient calculation with image domain PGD. We have added these results to Table 10 in the appendix. On comparing the results to Table 1, we see the spectral component does significantly contribute to VISOR++ performance. Momentum: We add ablations to VISOR++ by replacing double momentum from the VISOR++ Algorithm (Algorithm 1) in the appendix in the form of no momentum (Table 11) and single momentum (Table 12). By comparing the results from these tables we can observe that momentum is indeed necessary to obtain optimal performance. The performance of VISOR++ shows the following relation: Double momentum > Single Momentum > No momentum.
>
>
> __Comparison with other techniques (ASTRA, SteerVLM)__ We appreciate the reviewer’s suggestion to compare against ASTRA and SteerVLM, but both operate in fundamentally different regimes from VISOR++. ASTRA is an activation-space defense that alters internal representations to counter adversarial image jailbreaks, whereas VISOR++ is an input-space steering method that emulates steering vectors via optimized images for behavioral control across diverse, including closed, models. Given the differing objectives (defense vs. controllability), access assumptions (internal vs. external), and intervention layers (activations vs. inputs), ASTRA is not directly comparable—though the two approaches could be complementary, e.g., VISOR++ could emulate ASTRA-style steering in input space.
>
> SteerVLM, like ASTRA, performs internal activation manipulation and requires model access to insert steering modules. In contrast, VISOR++ provides a model-agnostic, input-based mechanism that approximates such effects without internal access, targeting a distinct deployment setting. Therefore, comparisons to ASTRA or SteerVLM would not be directly meaningful for evaluating VISOR++, which focuses on input-space, cross-model steering rather than internal fine-grained control.
>
> __Typos:__ We have fixed the inconsistencies between LLM/LLMs and VLM/VLMs, and the description of the MMLU dataset. We appreciate your keen eye in helping us find these typos! We have added citations for all the used models (e.g.,LLaVA, Mistral, Qwen2-VL, LLaVA-NeXT, Llama, etc. as well)
>
> __VISOR++ images visualization__ We have added VISOR++ images to the paper in Figure 3 and Figure 4 in the appendix.
>
> __Effectiveness of PGD:__ We referred to PGD being effective in terms of finding a successful steering image for a single model as shown in Table 1. However, as we show in our ablations, it is not as effective when training a single image over an ensemble of models.

---

> > ### Comment · Reviewer_EZDE · 2025-11-27
> >
> > Thank you for the clarifications and updates. While the additional ablations are helpful, I still have some remaining concerns regarding the justification and quantitative contribution of the dual-momentum optimization. It would also be very helpful if the updated content in the revised PDF could be clearly highlighted to improve readability. For now, I prefer to keep my score.

---

> > > ### Author Response · Authors · 2025-11-30
> > >
> > > We’re glad you found the ablations helpful. We have now included Figure 5 that clearly highlights the incremental gains of each of the components used in VISOR++ and justifies the quantitative contribution of dual momentum optimization used in VISOR++ by showing it offers the greatest steering range.

---

### Official Review · Reviewer_AUEq · 2025-10-31

**Soundness:** 3
**Presentation:** 2
**Contribution:** 3
**Rating:** 6
**Confidence:** 3

**Summary:**

This paper introduces VISOR++, a visual input–based steering method for large vision–language models (VLMs). Instead of modifying internal activations, VISOR++ learns a single optimized image that reproduces the effects of steering vectors, steering model behavior along safety dimensions like refusal, sycophancy, and survival instinct. The approach shows strong parity with activation-based steering while being compatible with API-based, closed-source models.

**Strengths:**

(1) Clever idea — turning activation steering into an input-space optimization makes the method deployable even when model internals are inaccessible.

(2) The algorithm is well-engineered, using differentiable preprocessing and spectral augmentation for cross-model robustness.

(3) Experiments are thorough, including both open-source and API-based models, and show that VISOR++ doesn’t degrade unrelated task performance.

**Weaknesses:**

(1) While the concept is novel, the empirical gains over activation steering are modest; VISOR++ mostly matches, not surpasses, existing methods.

(2) The paper treats the steering image as universal but doesn’t explore why it generalizes. There’s no mechanistic or interpretive analysis linking image features to behavioral shifts.

(3) Transfer results to unseen models (e.g., GPT-4V, Claude Sonnet) are very small and mostly directional, so the claim of “universal” transferability feels overstated.

**Questions:**

(1) How stable are these steering effects over different prompts or seeds? Is the learned image really robust, or do small text changes break the effect?

(2) The paper focuses on three behaviors, could the same idea work for other safety directions (e.g., toxicity, helpfulness), or is it limited to the chosen datasets?

(3) Have the authors looked at what the optimized images actually look like? Any visible structure or pattern that hints at how they interact with the vision encoder?

(4) Since VISOR++ depends on adversarial optimization, how reproducible are the results across runs — do different random inits yield similar steering effects?

---

> ### Author Response · Authors · 2025-11-23
>
> Thank you for your review. We appreciate your feedback on our paper. We are happy to hear that you find our idea clever, our method well-engineered, and our experiments thorough.
>
> __Empirical gains equivalent to steering vectors:__ As correctly identified by the reviewer, our empirical results indeed follow that of steering vectors. However, this is not an artifact, but the goal of VISOR++ which is to mimic the performance of steering vectors but in a less invasive manner. Since our method is purely an input space based intervention, we can transfer the steering technique from being a supply chain approach to input-based. Furthermore, since steering vectors are, by definition, model specific, they are more limiting, whereas in our approach, we show that for at least 2 VLMs with different architectures, we could train a single image to steer both the models.
>
> __Justification for universality:__ We refer to the common comment where we clarify how we utilize the terms “universality” and “generalizability”. We only refer to the images being universal in that they can be applied for a range of prompts for that particular behavior such as sycophancy, refusal, survival.
>
> __VISOR robustness to changes:__ We use a diverse set of prompts during training in order to train a single VISOR++ image. We have added a sample of the prompts from each dataset in Table 5 in the appendix. These prompts differ sufficiently in length, structure, and expected responses to make the learned image robust to changes in text prompts. Also, in our evaluation, we report results that are across a set of test prompts rather than a single prompt for that behavior type. VISOR++ also gives images that are robust to minor changes in training such as random seed. We have included images from other runs in Figure 4, in which we show that the images all have comparable steering performance.
>
> __Visible patterns in images:__ We have included the images corresponding to the results in the paper in appendix in Figure 4. We do not see any obvious pattern in the images. Our conjecture is that since the method aims to induce behavioral concepts rather than specific visual concepts, the patterns observed are not visually discernible.

---

### Official Review · Reviewer_3gJx · 2025-11-01

**Soundness:** 3
**Presentation:** 3
**Contribution:** 3
**Rating:** 6
**Confidence:** 3

**Summary:**

Existing behavioral control methods have limited transferability across different VLM architectures. To address this, the authors propose VISOR++, a method that enables behavioral steering using optimized visual inputs, without requiring access to model internals. They show that VISOR++ can reliably influence model behavior along three axes: refusal, sycophancy, and survival instinct. Also, they show that VISOR++ exhibits generalization to unseen models while having minimal impact on unrelated task performance.

**Strengths:**

- The motivation for this work is clear.
- The authors propose VISOR++, a technique that reframes behavioral steering from a latent-space intervention into an input-space control problem.
- VISOR++ avoids the need for white-box model access, making it applicable even to closed or API-served VLMs.

**Weaknesses:**

- VISOR++ is evaluated using only two models for image optimization, which makes it difficult to assess how broadly the method generalizes. Moreover, the behavioral shifts reported in Table 2 are often quite small, raising questions about the practical significance of the approach.
- The paper does not provide a clear explanation for why VISOR++ works. Could you share your intuition or analysis on the method?

**Questions:**

- Figure 1 is somewhat confusing: both example responses appear identical, making it unclear what behavioral difference the figure is intended to highlight. Clarifying this would improve readability.
- It is also unclear whether VISOR++ operates orthogonally to common image augmentations. For instance, does cropping, resizing, or compression diminish the steering effect?
- Could you add an experiment where steering vectors from one VLM are transferred directly to another VLM as a baseline comparison to see their generalization?

---

> ### Author Response · Authors · 2025-11-23
>
> Thank you for your feedback. We are glad to know our motivation came across well. We believe it is an important issue in the security of deployed VLMs.
>
> __Practical significance of VISOR++__ The key practical significance of VISOR++ is that it removes 2 constraints that limit the application of steering vectors: a) requiring access to model internals at runtime, b) being strictly applicable to one model only. On the other hand, we have shown in our attempts that one can successfully train a single image that can steer 2 models with different architectures. Application of VISOR++ images do not require access to model internals at runtime. We also showed that a naive PGD technique to train across the ensemble works poorly in achieving steering for either of the models. While we observe minimal but directional steering for unseen models, we are limited by compute and time constraints (14 hrs/image on Nvidia L40S for 2 models) when it comes to having more models in the ensemble. However, we firmly believe this is not a limitation of the proposed technique but a resource-limited challenge. While this is a limitation for research purposes, the practical advantage of this technique is that the steering images will have to be computed only once across a large ensemble and applied at runtime without requiring model access.
>
> __Intuition for VISOR++:__ The main idea motivating VISOR++ is the activation(s) that are achieved by performing activation addition can be replicated using input images if optimized for that exact objective. This is more easily possible by the richness of representation available in the input image space to help induce the steering level activations, when compared to text space. For instance, one could technically compute GCG suffixes to replicate steering vectors, but due to the discrete nature of text inputs, but the optimization will be a lot slower.
>
> __Figure 1:__ We have redrawn Figure 1 to improve clarity.
>
> __Effects of image augmentation:__ This is an excellent point! When computing the common steering image across models, we utilize a 384x384 image. Each model will resize and normalize it using its own preprocessing pipeline. E.g., LLaVA 1.5 resizes images to 336x336 during pre-processing. Furthermore, we already introduce a slight mismatch during the steering image computation since we reimplemented the differentiable preprocessing approximation using a different interpolation technique (bilinear) from the huggingface version (bicubic). Even with these above differences, the final VISOR++ images computed can still hold their performance during the inference stage when using standard preprocessing that includes rescaling and normalization.
>
> __Direct transferability results:__ We ran the suggested experiments for understanding if model specific VISOR++ images would transfer to the other model. These results have been added to the paper in Table 13 in the appendix. We observe that when VISOR++ images are trained on only 1 model, they do not transfer at all to the other model, and in some cases, results in opposite steering effects.

---

> > ### Comment · Reviewer_3gJx · 2025-11-27
> >
> > Thank you for your detailed response.
> >
> > I understand the difficulty regarding the computational limit; however, it remains difficult to claim that the method generalizes broadly without supporting experiments. Since this is one of the main claims of the work, it continues to remain as a concern. Aside from this point, I am satisfied with the clarification and will maintain my score of weak accept.

---

> > > ### Author Response · Authors · 2025-11-30
> > >
> > > We would like to emphasize our key contribution: our work is the __first to demonstrate that steering vectors can be emulated through visual inputs alone__. We view this as the main contribution as this __transforms a well-known supply chain technique to a practical approach__ independent of post-hoc modifications such as steering vectors, thus applicable to scenarios where models are served using API endpoints. While black box transferability of VISOR++ images is our secondary goal, VISOR++ still achieves a __7% point negative steering__ for anti-survival and sycophancy cases for the __closed-access GPT-4 Turbo__ which is quite significant given our limited ensemble size due to severe compute and time constraints.

---

### Author Response · Authors · 2025-11-23
**Addressing common concerns about the use of "Universal" and "Transferable"**

We thank all reviewers for their thoughtful feedback. We are pleased that our motivating problem, method novelty, engineering quality, and thorough experiments were well received. We also appreciate the recognition that VISOR++ is both practical and deployment agnostic which were core goals of our work. Before addressing individual reviewer comments, we clarify the our usage of “universality” and “transferability” below:

__Universality:__ The image is universal if it can be utilized to steer any accompanying text from the prompt distribution for a given model. In our approach, both when we utilize PGD as well as when we utilize the dual momentum with spectral domain augmentation, we train a universal image either for a single model or across models such that the image achieves steering across a range of prompts for that behavioral task. We achieve this using the well-known Expection-over-Transformation approach of training the image along with a diverse set of prompts from the behavioral steering task such as sycophancy, refusal, etc.

__Transferability:__ The image is transferable if it can induce behavioral steering for models that were not encountered when computing the image. In our work, we trained “ensemble VISOR++” images that can steer two models at the same time. Given our resource constraints both in terms of compute resources as well as time taken to train a steering image, we were only able to utilize 2 models in our ensemble. As a result, while we were able to show promise of directional steering effects in Table 2 for unseen models, we do not claim that the VISOR++ images obtained from a model ensemble size of 2 are fully transferable to unseen models.

We have changed the phrase "Universal VISOR++ image" to "Ensemble VISOR++ image" in the paper to help make the distinction between universality and transferability of VISOR++ clearer.

---

### Author Response · Authors · 2025-11-30

We thank the reviewers for the feedback to improve and clarify the main contributions in our paper. We have provided __all of the ablation studies__ requested by Reviewer 3gJx (Table 13), Reviewer AUEq (Figure 4), __as well as the additional analyses__ noted by Reviewer EZDE3 (Table 10, Table 11, Table 12, Figure 5) and justified the need for the techniques outlined to achieve the maximum steering range. We would like to take this opportunity to clearly restate the core contribution of our work:

- Our work is __the first to demonstrate that steering vectors can be emulated through visual inputs alone__. We view this as the main contribution as this __transforms a well-known supply chain technique to a practical approach__ independent of post-hoc modifications such as steering vectors, thus applicable to scenarios where models are served using API endpoints.

- Since steering images do not carry the same HARD requirement of being model-specific as would be the case for steering vectors, we also demonstrate that one could train a single steering image to steer two vision-language models of very different architectures. __Through ablations, we demonstrate various ways to achieve this effectively and show that VISOR++ is the best way to achieve image based steering (Figure 5)__. Furthermore, prior work leveraging spectral augmentation and dual momentum has shown success in computing transferable images for large ensembles under limited vision encoder only model settings, adding credibility to this effort.
- We observe directional transfer to new models outside of the ensemble (Table 2). Among them, the better transfer was achieved to models from the same family as that of the ensemble. We even achieved a __7% point negative steering__ for anti-survival and sycophancy cases for the __closed-access GPT-4 Turbo__ which is quite significant given our limited ensemble size due to severe compute and time constraints.
- We want to emphasize that our work opens up the idea of using images to craft transferable steering images. Transferable adversarial images for VLMs is very much an actively studied problem, so as better solutions emerge, one could easily apply the same tools to improve the transferability of VISOR++.

---

### Meta-Review · Area_Chair_wEA2 · 2026-01-12

**Summary:**

This paper introduces VISOR++, a method for steering the behavior of vision–language models using optimized visual inputs alone, without requiring access to model internals or runtime steering vectors. The authors show that a single universal image can induce targeted behavioral shifts (e.g., refusal, sycophancy, survival instinct) across multiple open and closed VLMs, achieving performance comparable to activation-based steering methods. The approach is demonstrated on several models and shows partial transferability to unseen models while largely preserving performance on unrelated tasks.

During the rebuttal, the authors addressed some of the reviewers’ concerns, but one major issue remains, as pointed out by Reviewer EZDE: the justification and quantitative contribution of the dual-momentum optimization. While the authors provided Figure 5 to illustrate the quantitative impact of this component, the underlying justification and insight into why it works are still missing. In addition, as highlighted in the authors’ response, “we referred to PGD being effective in terms of finding a successful steering image for a single model, as shown in Table 1; however, as we show in our ablations, it is not as effective when training a single image over an ensemble of models.” It is difficult to identify which ablation study sufficiently supports the claim that PGD optimization is less effective in the ensemble setting. This therefore limits the necessity of the dual-momentum optimization.

We agree that the novelty of generating a single image to steer behavior across both open and closed VLMs is good, and we hope the authors take the reviewers’ feedback into account to further strengthen the work. Taken together, however, we recommend rejecting this work in its current form.

**Reviewer Concerns:**

Reviewer 3gJx’s concern about the lack of evidence supporting the claim of broad generalization remains unresolved.

Reviewer AUEq’s concerns that the steering image is treated as universal without sufficient explanation of why it generalizes, and that transfer results to unseen models are limited and mostly directional, have not been addressed. In addition, their questions about whether the approach extends to other behaviors (e.g., toxicity or helpfulness) and about the reproducibility of the adversarial optimization across runs also remain unanswered.

Reviewer EZDE’s concern regarding the justification and quantitative contribution of the dual-momentum optimization remains.

Reviewer w7Jw’s own concerns have been addressed, but this reviewer agrees with Reviewer EZDE’s remaining concern.

**Reviewer Scores:**

Reviewer 3gJx (score: 6), Reviewer w7Jw (score: 6) and Reviewer EZDE (score: 2) have responded the reviewers and decide to maintain their score.

Reviewer AUEq would potentially remain the score 6 as the concerns are only partially addressed.

---

### Decision · Program_Chairs · 2026-01-26

Reject